# Testing Textural Information Base on LiDAR and Hyperspectral Data for Mapping Wetland Vegetation: A Case Study of Warta River Mouth National Park (Poland)

Anna Jarocińska [1], Jan Niedzielko [2], Dominik Kopeć [2,3,*], Justyna Wylazłowska [2], Bozhena Omelianska [1] and Jakub Charyton [2]

1   Department of Geoinformatics, Cartography and Remote Sensing, Chair of Geomatics and Information Systems, Faculty of Geography and Regional Studies, University of Warsaw, 00-927 Warsaw, Poland; ajarocinska@uw.edu.pl (A.J.); b.omelianska@student.uw.edu.pl (B.O.)
2   MGGP Aero sp. z o.o., 33-100 Tarnów, Poland; jniedzielko@mggpaero.com (J.N.); jwylazlowska@mggpaero.com (J.W.); jcharyton@mggpaero.com (J.C.)
3   Department of Biogeography, Paleoecology and Nature Conservation, Faculty of Biology and Environmental Protection, University of Lodz, 90-237 Łódź, Poland
*   Correspondence: dominik.kopec@biol.uni.lodz.pl

**Abstract:** One of the key issues in wetland monitoring is mapping vegetation. Remote sensing and machine learning are used to acquire vegetation maps, which, due to the development of sensors and data processing and analysis methods, have increasingly high accuracy. The objectives of this study were to test: (i) which of the textural information (TI) features have the highest information potential for identifying wetland communities; and (ii) whether the use of TI improves the accuracy of wetland communities mapping using hyperspectral (HS) and Airborne Laser Scanning (ALS) data. The analysis indicated that the mean and entropy features of the Gray Level Co-occurrence Matrix had the highest potential to differentiate between various wetland communities. Adding these features to the dataset resulted in a small increase (0.005) in average F1 accuracy based on HS data and 0.011 for HS and ALS scenarios in wetland communities classification, and adding TI improved the delineation of patch boundaries. A higher increase was noted for forest and scrub vegetation (by 0.019 for the HS scenario and 0.022 for the HS and ALS scenario) and rushes (only for the HS and ALS scenario 0.017). It can be concluded that it is reasonable to use textural information for mapping wetland communities, especially for areas with a high proportion of scrub and forest and rushes vegetation included in the analysis.

**Keywords:** ALS; machine learning; classification; data fusion; floodplain; Permutation Importance; CatBoost; HySpex; GLCM

## 1. Introduction

Wetlands perform various important functions—for example, they regulate global and local climate [1], they are a location for rare species, and their proper functioning contributes to the protection of biodiversity [2]. Unfortunately, these areas are influenced by climate change, especially the rise in temperature [3]. Any disturbances in the climate lead to changes in hydrology and, as a consequence, changes in vegetation [1], so vegetation monitoring can therefore detect changes in the functioning of wetlands [4].

Vegetation mapping over large areas (such as national parks) with limited accessibility can be difficult using conventional methods. The use of remote sensing techniques can be a good and recommended solution [5] because it allows for a significant reduction in field measurements and it enhances the objectivity and comparability of results [4]. In addition, the intensively developed machine learning (ML) algorithms currently allow different types of data, such as optical and ALS, to be combined and processed [6]. ML has high computational capabilities and adapts to the input data, which is why it can be a

good tool for processing remote sensing data. One of the newly introduced algorithms is CatBoost, which was successfully used in the classification of remote-sensing images [7–9]. The combination of remote sensing data and ML algorithms gives good results in vegetation classification and thus may be useful in classifying areas such as wetlands [10]. On the other hand, the identification of diverse, natural vegetation based on remote sensing data may not be precise enough. That is why there are studies to test the new techniques that could improve ML results and better identify such vegetation using remote sensing methods [4,11].

One of the main purposes of vegetation identification is its application in nature conservation. However, the utility of this purpose is dependent on the high accuracy of the acquired map, so it is important to test and implement methods that improve this. The accuracy depends on many factors: the input dataset—types of data and used resolutions, reference datasets, preprocessing procedures and classification procedure. One way to improve the mapping results is to use relevant data as input; an example could be the fusion of different spectral resolutions [11]. The spectral reflectance for vegetation is very diverse, which makes identification by remote sensing techniques difficult. This problem is particularly important in the case of diverse vegetation with high internal variability, such as wetlands [4], so one of the solutions is to use data with a high spatial and spectral resolution—for example, aerial hyperspectral data (HS) [12–14]. In addition, one of the most successful data fusions used to improve mapping quality is a combination of HS and Airborne Laser Scanning (ALS) data [15]. For Natura 2000 habitat mapping, using the fusion of aerial HySpex images and ALS data, the F1 value for the Natura 2000 habitat varied from 0.923 to 0.953 and was dependent on time and dataset. On the other hand, even based on this combination of data with high resolutions, it is still not always possible to correctly identify all habitats [10,16].

The natural vegetation is varied within the patches (has a high $\alpha$-diversity), often creating distinctive textures. On the other hand, the classifications using raster data are mainly conducted on a pixel level, and each pixel is identified independently from its neighbour. Based on the vegetation structure, it is worth using not only spectral features for identification but also those that take into account the pixel's neighbourhood. One solution is to use textural information (TI) to estimate the internal complexity of the identified classes, as well as differences in community patches. So far, TI has been most often used to identify forest vegetation. Mangroves were classified based on WorldView-3 data, including TI, with an overall accuracy (OA) of 0.94 [17]. In this case, the features of homogeneity, contrast, entropy and correlation turned out to be significant. The Gray Level Co-occurrence Matrix (GLCM) features were also used to classify forests, which allowed OA = 0.92 to be achieved [18]. Based on data from the HyMap HS image, the TIs were calculated, and tree species were classified [19]. Adding TI increased OA by 0.05 to 0.61. Adding TI also improves the results of classification for crops [20]. However, other authors have not found an increase in the accuracy of *Phragmites australis* identification when adding TI [21].

There are few studies evaluating the effectiveness of additional TIs for the identification of non-forest natural and semi-natural communities. Based on data from WorldView-3, TI was calculated, and this combination resulted in an OA of 0.91 for invasive weed classification [22]. Moreover, the chosen TI features (mean, entropy, dissimilarity) calculated from IKONOS images increased the accuracy of the classification of sub-Antarctic plant communities by 0.06 [23]. However, there are no studies that check whether the TI calculated on HS and ALS data improved the classification results of wetland communities.

Very few studies were dedicated to the analysis of the usefulness of TI for identifying natural and semi-natural non-forest communities, and none of these was dedicated to wetland identification. TI brings new information, which is useful for class differentiation. At the same time, adding more layers to the classification increases processing time. This is especially important when using high-dimensional data such as HS. In this case, a significant part of the analysis focuses on feature reduction. Different feature selection techniques can be used to reduce the volume of data—for example, Permutation Importance [24] or

Recursive Feature Elimination [25]. Therefore, adding additional layers is only justified if they increase the accuracy of identification. To properly determine the significance of layers, it is necessary to compare the mapping accuracy on the same dataset with and without TI.

The aim of this study was to test the applicability of selected TI using Permutation Importance (PI) for mapping wetland communities using HS and ALS data. The following research questions were defined:

(i) Which TI calculated from HS and ALS has the greatest potential for wetland communities mapping?

(ii) Does the TI calculated from the HS, or the HS and ALS data, improve the accuracy of the wetland communities mapping?

(iii) For which plant communities does the use of TI increase the classification accuracy?

The analyses were divided into two experiments: (1) identification of potentially influential TIs based on Permutation Importance and (2) comparison of classification results with and without TIs included in the input dataset.

## 2. Study Area and Object of Research

The study was conducted in the Warta River Mouth National Park, located in western Poland, in the mesoregions of the Gorzow Basin and Freienwalde Basin—a fragment of the Toruń-Eberswalde Ice Marginal Valley [26]. The survey covered the entire park and part of its buffer zone—a total of 110.8 km². Experiment 1 was carried out on a selected area of 15.6 km², representative of the park's vegetation, while Experiment 2 was carried out on the entire analysed area (Figure 1).

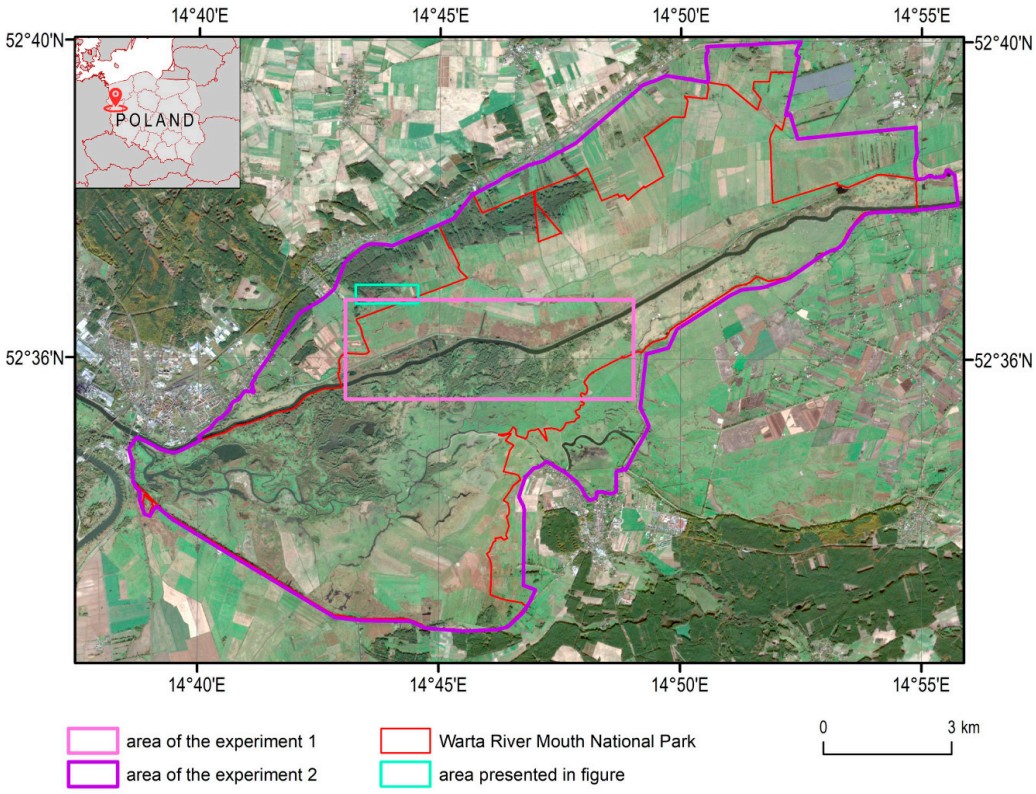

**Figure 1.** The study area, Warta River Mouth National Park.

The Warta River floodplains are of key importance for wetland plant communities and, associated with them, many animal species, especially birds. For this reason, the analysed area has been placed under legal protection, first as the Słońsk nature reserve (in 1977) and, since 2001, as the Warta River Mouth National Park. In 1984, the research area was included in the areas of the Ramsar Convention. In 2007, the analysed area became part of the European Natura 2000 network as PLC080001 "Ujście Warty" [27]. The largest

area of the park is covered by Holocene peats up to several metres thick. Small areas are also occupied by rivers, mud and sands [28,29]. The relief and true altitude are not very diverse. The lowest point of the analysed area is at 9.6 m msl, the highest at 20.6 m msl, and the average altitude is 11.7 m msl.

The park has a complex hydrological network. Its axis is the Warta River, which flows through the park's territory in a straight, regulated channel, and just outside the park's borders, it flows into the Oder River. The other important river in the park is the slow-flowing, meandering Postomia River. The Warta and the flood control dike running along its right bank divide the park into two distinctly different areas. North of the Warta is the higher-lying Polder Północny (North Polder), cut by numerous ditches and canals from which water is pumped into the Warta. The water level there is relatively stable. South of the Warta riverbed is a flat, extensive floodplain (Figure 2). This area can remain underwater for part of the year, and water level fluctuations here can reach up to 4 metres per year [30]. It should be noted that the flooding of the park's area is caused by backwaters from the Oder River and waters carried by the Warta River from outside the park. Precipitation in the area is less than the national average and amounts to 500–550 mm per year, of which about half falls in May–August. The winter period is the poorest for precipitation, and the duration of snow cover does not exceed 25 days. Average annual air temperatures vary from 7 to 9 °C [30,31].

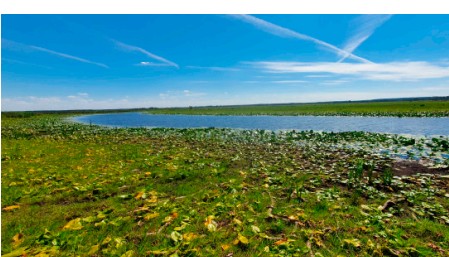 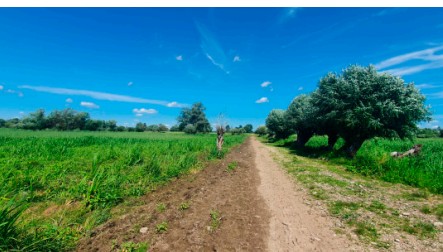 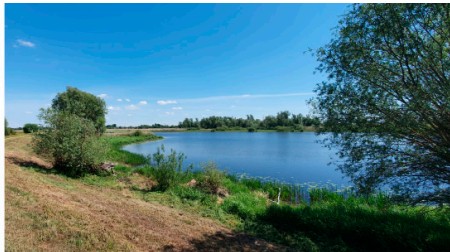

**Figure 2.** The photographs of study area.

Until the 18th century, the Warta flowed in many channels and branches in the analysed area, creating a complex hydrological network in a wide, swampy valley overgrown with riparian forests. In the 18th century, the area was drained and deforested. Livestock grazing and hay harvesting began in the resulting meadows and pastures. As a result of these activities, most of the park is now non-forested. The vegetation is dominated by reeds, rushes, meadows and pastures. On muddy water banks, communities of annual plants develop. The original vegetation is alluded to by willow thickets and small-area forests dominated by alder or willow. There are several Natura 2000 habitats in the park, including some of European Union priority importance: 3130, 3150, 3260, 3270, 6120 (priority), 6210 (priority), 6430, 6440, 6510, 9170, 91E0 (priority), 91F0 [27].

## 3. Materials and Methods

### 3.1. Aerial Data

The aerial dataset was acquired on 22 June 2020. The flight was conducted with the use of a multisensor platform consisting of a RIEGL LiDAR VQ-780II scanner and HySpex hyperspectral scanners: VNIR-1800 (400–1000 nm spectral range) and SWIR-384 (930–2500 nm) [32]. The platform was slightly modified by adding a second SWIR-384 scanner to increase the spatial resolution of images in the SWIR range. Table 1 describes all the sensors used and the basic settings of the flight parameters. For sensor specification details, see the producer's materials: Norsk Elektro Optikk AS, Skedsmokorset, Norway, for HySpex, and RIEGL Laser Measurement Systems GmbH, Horn, Austria, for VQ-780II [33,34]. Flights were conducted at an altitude of 1.3 km with a minimum sun elevation angle of 30°.

**Table 1.** Characteristics of the sensors used in the study with the main parameters of data acquisition.

| Sensor | Producer | Spatial Resolution | FOV | Side Overlap |
|---|---|---|---|---|
| HySpex VNIR-1800 | NEO | 1 m GSD | 34 | 30 |
| HySpex SWIR-384 x2 | NEO | 1 m GSD | $2 \times 16$ (~1 overlap) | ~30 |
| VQ-780II | Riegl | 7.6 point/m$^2$ (>15 with overlap) | 60 | 50 |

*3.2. Image Preprocessing*

Hyperspectral data were parametrically geocoded in PARGE software (ReSe—Remote Sensing Applications, Wil, Switzerland) [35] using flight navigation data and a camera sensor model. Images were orthorectified on Digital Surface Model created from ALS data. Raw data values were converted to at-sensor radiance (W·nm$^{-1}$·sr$^{-1}$·m$^{-2}$) in the software supplied by the equipment manufacturer (Norsk Elektro Optikk AS, Skedsmokorset, Norway) [33]. Images from VNIR and the two SWIR scanners were combined geometrically into a single hyperspectral data cube, setting split wavelength at 935 nm and trimming the rows to a spatial range of SWIR imagery. Atmospheric compensation was made using ATCOR4 software (ReSe Applications GmbH, Wil, Switzerland) [36] using flat terrain topography, variable water vapour, visibility estimation, and rural aerosol type. Bands of wavelengths longer than 2.35 µm were removed due to a high noise level. A Savitzky–Golay filter with a 6-band window side was applied to polish the spectra. The images were mosaicked to the middle of overlapping areas between the flight lines.

The ALS point cloud orientation was made using RiProcess software (RIEGL Laser Measurement Systems GmbH, Horn, Austria) [34]. Full waveform decomposition into a point cloud was done using RiAnalyze software (RIEGL Laser Measurement Systems GmbH, Horn, Austria) [37]. Point cloud classification was automatically pre-classified in TerraSolid software (Terrasolid Ltd., Helsinki, Finland) [38]. After that, the point cloud was manually classified into ASPRS standard classes.

As a part of preprocessing, the Minimum Noise Fraction (MNF) transformation was calculated from the hyperspectral images, and 30 first bands were chosen for further analysis. As additional information, seven spectral indices (SI) were calculated based on the previous experiments and the literature [25]: Anthocyanin Reflectance Index 2, Carotenoid Reflectance Index 1, Clay Minerals Ratio, Iron Oxide Ratio, Normalized Difference Nitrogen Index, Red Green Ratio Index, WorldView Water Index. The indices were calculated using ENVI software (Harris Geospatial Solutions, Broomfield, CO, USA) and the "Spectral Indices" tool [39]. The reference to spectral indices can be found in the tool's online manual (ENVI Spectral Indices tool). A full list of SI is available in Table A1.

A classified point cloud was processed from ALS data to raster layers in order to be used as features in classification. The digital terrain model (DTM) was calculated in the TerraSolid software package [40] using the linear interpolation method from the points classified as ground. Statistical metrics calculated in a 1 m spatial pixel cell are referred to as ALS features (ALSF). These were calculated in lidR [41]. The lidR library can calculate multiple statistical measures (mean, percentile, standard deviation, disparity, etc.) that describe vegetation properties—for example, height, reflectance and density. Apart from using the standard metrics delivered by the library, a set of custom metrics was defined and calculated for the study area. A full list of ALSFs is available in Table A2. A group of topographic indices (TOPO) was calculated based on the terrain model using SAGA software [42] (Table A3).

*3.3. Reference Data*

Together with the acquisition of aerial data, ground reference data were acquired. The time of data acquisition (both aerial and terrestrial) was designed to capture the full development of plant communities located in the study area. The field measurements were carried out in the period 8 June 2020–27 August 2020. The main goal in acquiring the

ground reference data was to collect a set representing the variability of all the communities in the study area.

A total of 1608 reference polygons were collected, including 1475 representing plant communities and 133 polygons for other land cover elements—for example, areas without vegetation and surface water (Table 2). Reference polygons were circles with a defined radius and phytosociological affiliation. The radius ranged from 1 to 5 m, depending on the type of plant community—aquatic and non-forest vegetation 1 to 3 m, shrub and forest vegetation 3 to 5 m. Phytosociological affiliation was determined by the species composition and plant proportions in the patch. The coordinates of the centre of the circle were recorded in the field using a Trimble Catalyst DA1 GNSS system [43], with a measurement accuracy of 1 m, and a mobile application for GIS data collection—Mapit [44].

**Table 2.** Plant communities are represented in the reference botanical ground data.

| Experiment 1 | | Experiment 2 | | | |
|---|---|---|---|---|---|
| Class Name | Ref. Polygons [1] | Class Name—Syntaxonomic Units | Class Description | Vertical Structure (Plant Dominants) | Ref. Polygons [1] |
| Aquatic vegetation | 10/232 | *Lemnetea* and *Potametea* | Aquatic macrophyte vegetation from *Cl. Lemnetea minoris* and *Cl. Potametea* | Underwater plants and plants on the water surface | 90/1971 |
| Rushes | 73/2084 | *Phalaridetum arundinaceae* | Marsh vegetation dominated by *Phalaris arundinaceae* | high perennials | 144/3885 |
| | | *Magnocaricion* | Marsh vegetation from *All. Magnocaricion* | high perennials | 96/2840 |
| | | *Phragmition* | Marsh vegetation from *All. Phragmition* | high perennials | 170/4603 |
| Annuals | 16/277 | *Isoëto-Nanojuncetea* | Amphibious short annual pioneer vegetation from *Cl. Isoeto-Nanojuncetea* | low annuals | 42/613 |
| | | *Bidentetea* | Annual pioneer nitrophilous vegetation from *Cl. Bidentetea tripartiti* | low annuals | 78/1867 |
| Meadows, grasslands and pastures | 33/1073 | *Trifolio-Agrostietalia* and *Plantaginetalia* | Pastures vegetation, periodically covered with flood water and vegetation of trodden surfaces from *O. Trifolio fragiferae-Agrostietalia stoloniferae* and *O. Plantaginetalia majoris* | low perennials | 110/3520 |
| | | *Molinietalia* | Wet meadows and nitrophilous perennials from *O. Molinietalia caeruleae* | low perennials | 190/6398 |
| | | *Arrhenatheretalia* | Lowland hay meadows from *O. Arrhenatheretalia* | low perennials | 54/1647 |
| | | *Koelerio-Corynephoretea* and *Festuco-Brometea* | Xeric sand semi-dry calcareous grasslands from *Cl. Koelerio glaucae-Corynephoretea canescentis* and *Cl. Festuco-Brometea* | low perennials | 107/3430 |
| Nitrophilous perennials | 23/392 | *Artemisietea* and *Epilobietea* | Nitrophilous perennials and shrubs from the *Cl. Artemisietea vulgaris* and *Cl. Epilobietea angustifolii* | high perennials | 190/4725 |
| Forests and shrubs | 29/1880 | *Salicetea purpureae* | Swamp forests and shrubs from *Cl. Salicetea purpureae* | shrubs and trees | 53/4208 |
| | | *Ribeso nigri-Alnetum* and *Alno-Ulmion* | Swamp forests from *Ass. Ribeso nigri-Alnetum* and *All. Alno-Ulmion* | trees | 56/4472 |
| | | *Salicetum pentandro-cinereae* | Shrub communities from *Ass. Salicetum pentandro-cinereae* | shrubs | 18/1055 |
| | | *Vaccinio-Piceetea* | Pine forests from *Cl. Vaccinio-Piceetea* and others communities with pine | trees | 20/880 |
| | | *Others wooded communities* | Wooded communities without syntaxonomic assignment | shrubs and trees | 82/2683 |
| Areas without vegetation | 10/171 | Areas without vegetation | Land areas without vegetation | no plants | 57/1591 |
| Surface water | 10/825 | Surface water | Surface water without aquatic macrophyte vegetation | no plants | 51/3553 |

[1] Number of reference polygons/Reference areas in m$^2$.

After completion of the field measurements, the collected reference polygons were combined into classes using syntaxonomic criteria, including vertical structure, resulting mainly from the type of plant dominants (Table 2) [45,46]. According to the criteria used, individual units may belong to different levels of the phytosociological hierarchy. The proposed division of classes reflects the actual diversity of the communities of the study area well. Reference polygons were distributed as evenly as possible, and the abundance of each class corresponded to the frequency of each plant community in the analysed area (Figure 3).

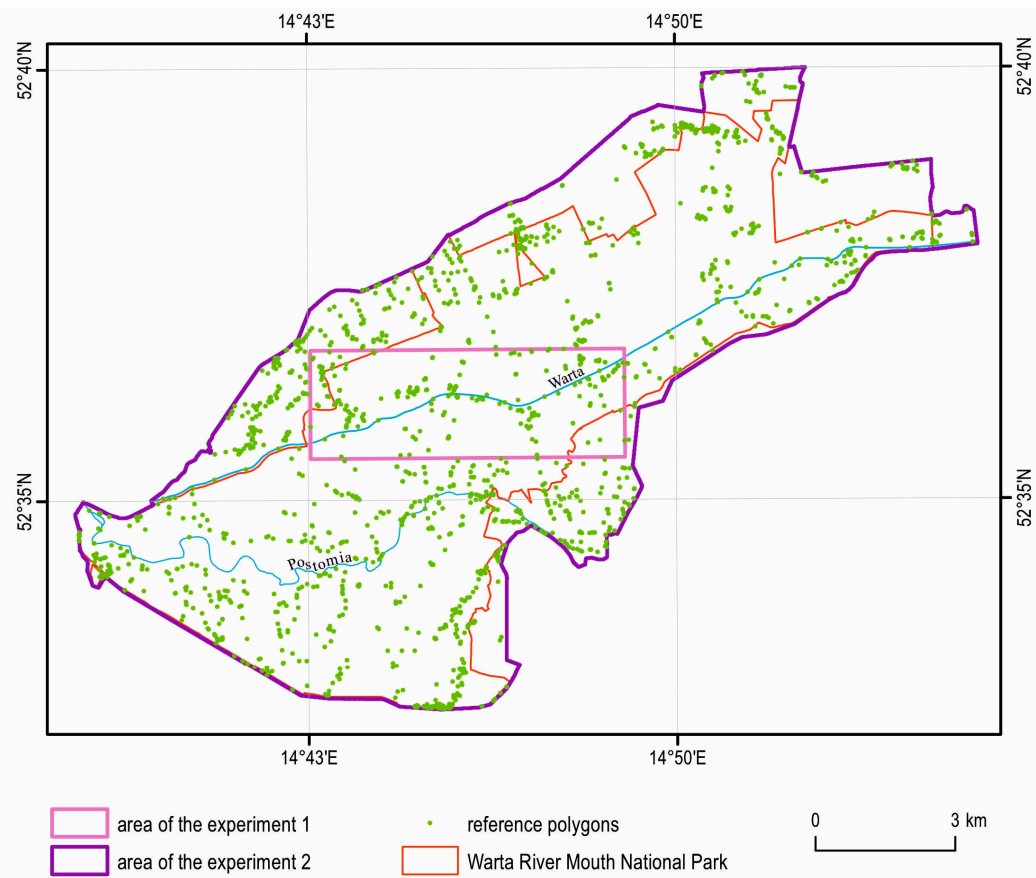

**Figure 3.** The distribution of reference polygons.

### 3.4. Data Analysis

The analyses were divided into two steps: identification of potentially influential TIs and comparison of classification with and without TIs included in the input dataset (Figure 4). To acquire information about the utility of TI, two experiments were performed. The first one aimed to optimize the analysis and find, using Permutation Importance, what TI could be influential in wetland communities mapping. The second experiment focused on the determination of the differences in classification accuracy between datasets without TI and with TI, separately, for HS images only and for HS and ALS data.

The results of both experiments were validated using reference data. The polygons were divided into training and validation data using stratified random sampling. Based on the validation data, an accuracy assessment was performed, where an F1 score for each class was acquired based on user and producer accuracy. Then, the average F1 for all classes (F1 macro score) was calculated, and this parameter was the basis of the analysis.

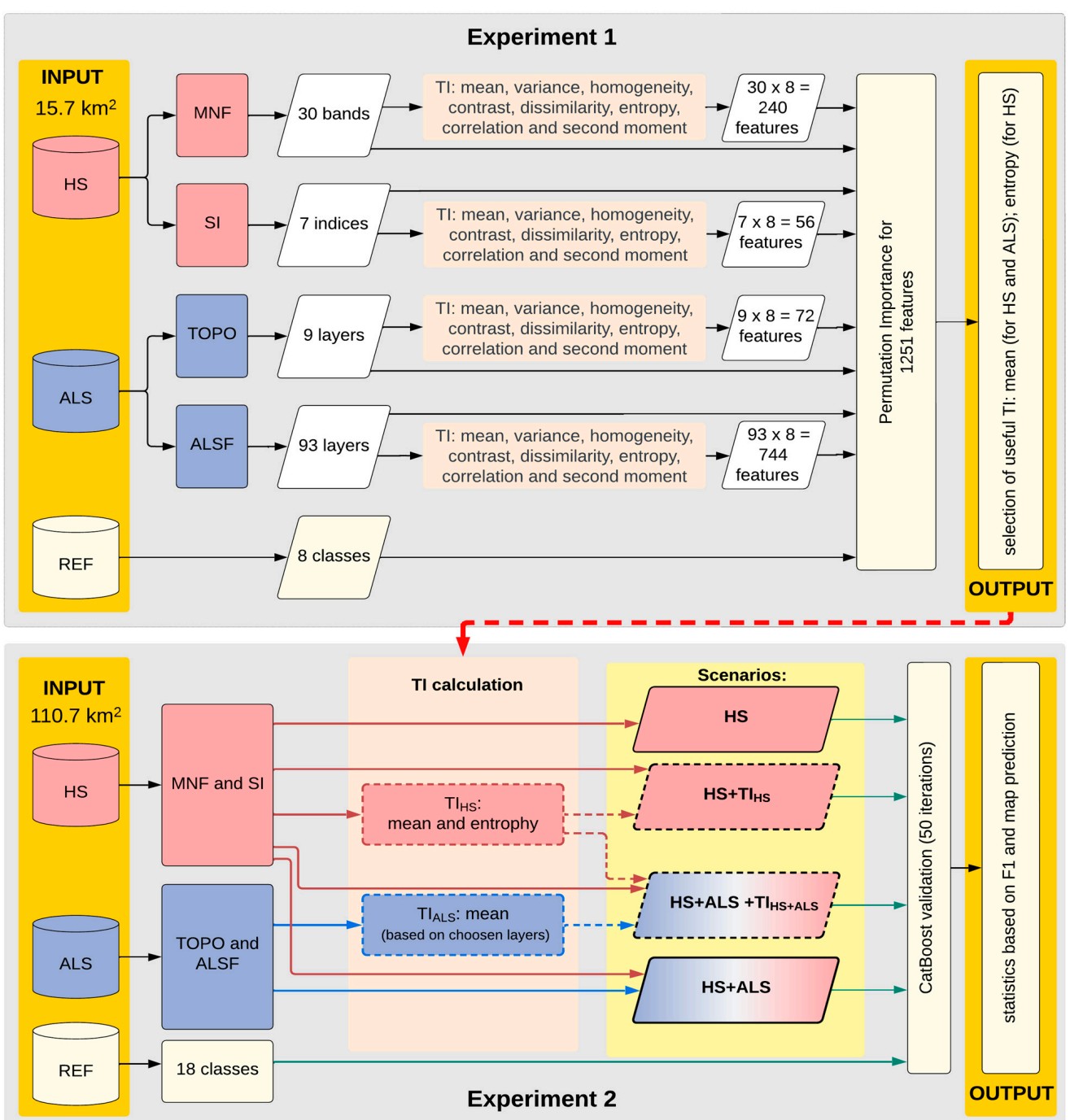

**Figure 4.** Research scheme of experiments 1 and 2; HS—hyperspectral data, ALS—Airborne Laser Scanning data, MNF—Minimum Noise Fraction, SI—spectral indices, TOPO—topographic indices, ALSF—ALS features, REF—reference polygons, TI—textural information, $TI_{HS}$—influential TI features based on HS data, $TI_{HS+ALS}$—influential TI features based on HS and ALS data, F1—accuracy calculated for classes in classification process.

### 3.4.1. The Determination of Influential TI (Experiment 1)

The first experiment was performed on a part of the study area. The area covers more than 15.7 km$^2$ (Figure 1). Additionally, image resizing reduced the amount of data and analysis time. The map legend was simplified by combining individual classes into higher syntaxonomic units (Table 2) to obtain a minimum of 10 polygons for individual classes.

The first step was to calculate TI GLCM (Table A4) [47]. Eight different texture features were calculated for each ALS and HS layer, with a kernel size of 3 per 3 pixels: mean, variance, homogeneity, contrast, dissimilarity, entropy, correlation and second moment (Table A5). Next, original HS and ALS bands were stacked with the calculated TI into one raster file with 1521 bands: 30 MNF bands, 240 TI based on MNF; 7 SI, 56 TI calculated using SI; 93 ALSF and 744 TI based on ALSF products; and 9 TOPO with 72 TI based on TOPO. For more information, see Section 3.2. Image Preprocessing.

To determine the TI influential in wetland communities mapping, a Permutation Importance (PI) analysis was conducted. The PI procedure is described as the decrease in a model's accuracy when one layer is randomly shuffled [48]. The drop in accuracy shows how much the model depends on the feature. The analysis was based on a mean decrease in the F1 macro score for each layer. The PI was performed using cross-validation (validation performed 10 times with the stratified sampling, with 90% calibration and 10% validation polygons). As a result, the value of PI was analysed for each layer for the 10 iterations. It was assumed that if the value was above 0, the layer potentially improved the classification accuracy. The TI was considered significant if the PI value was above 0 in at least 5 out of 10 iterations. A list of influential TI features was subsequently created: $TI_{HS}$ based on HS data and $TI_{HS+ALS}$ based on HS and ALS data.

### 3.4.2. Mapping of Wetland Communities with the Use of TI (Experiment 2)

The second experiment aimed to determine whether textured features significantly improve the classification accuracy of wetland communities mapping. The results from experiment 1 were used to determine classification data for the whole study area. Only products that were found to be influential in experiment 1 were calculated. To assess the usefulness of the TI based on the accuracy for 50 classification iterations, four scenarios were created: two for HS data and two for the combination of HS and ALS data. Two pairs were compared: for HS and HS + $TI_{HS}$ scenarios—to check if the texture layers improve classification results using only hyperspectral images, and for HS + ALS and HS + ALS + $TI_{HS+ALS}$ scenarios—to analyse whether the TI calculated for HS and ALS data improve classification results.

In the comparison CatBoost, classification was used. The classification process was conducted using the CatBoost library machine learning algorithms [49]. CatBoost is based on a gradient boosting technique that builds strong predictors by combining base (weaker) predictors through a greedy iterative procedure of fitting weak predictors with gradient descent. In the case of this library, base predictors are decision trees. The library uses symmetric trees with the same depth applied to each leaf node of the tree. The usage of symmetric trees reduces the overfitting of base predictors. Base predictors are iteratively added to the ensemble model.

New trees are built to approximate the gradients (error values) of the created model [50,51]. The target of this process is to find the optimal split points of the trees that correspond to the minimum loss function. CatBoost uses a biased pairwise gradient estimation technique to improve the speed of calculating gradient values, which are used to optimize the selected loss function. The CatBoost validation was performed 50 times on reference polygons with 50%/50% training/validation stratified random sampling on reference polygons.

For the accuracy assessment, F1 scores were calculated for each class and as an average from 50 iterations. The acquired results for four scenarios were analysed based on F1 values for each class and also for the mean F1 score. The mean values of F1 calculated from 50 classification iterations were compared in pairs: for HS and HS + $TI_{HS}$ scenarios and for HS + ALS and HS + ALS + $TI_{HS+ALS}$ scenarios. The data were tested for normal distribution, and based on the information, the t-test was performed to determine if there was a significant difference between the means of the two groups. The test was calculated for two datasets: the F1 values separately for each class and the average from all classes.

To compare the results, prediction maps were generated for each scenario. Due to the very large amount of data, a representative fragment of the study area was selected for visualization (Figure 1). Each of the 50 fitted models was used to produce a map, and the final class for each pixel was determined by majority voting. The results are presented in maps for each scenario.

## 4. Results

### 4.1. A Selection of TI Features Influential for Communities Identification (Experiment 1)

In PI, 1112 TI were analysed: 240 were calculated from MNF, 65 from SI, 72 from TOPO and 744 from ALSF. The goal of experiment 1 was to determine what TI could be considered influential in mapping wetland communities. This information was based on 10 iterations of PI. The results showed that 91 TI features were determined as potentially influential at least once: 34 layers based on ALSF, 18 based on TOPO, 38 based on MNF and 1 based on SI (Figures 5 and 6). Among the eight different types of TI (mean, variance, homogeneity, contrast, dissimilarity, entropy, correlation and second moment), the TI mean layers computed 16 from the HS and 26 from ALS layers, which were indicated as influential at least once by the PI algorithm, while the dissimilarity layer was the least influential: four from HS and eleven from ALS. Of the large number of TI features calculated from the ALSF, only 34 TIs calculated from the 24 ALSF layers were determined as influential.

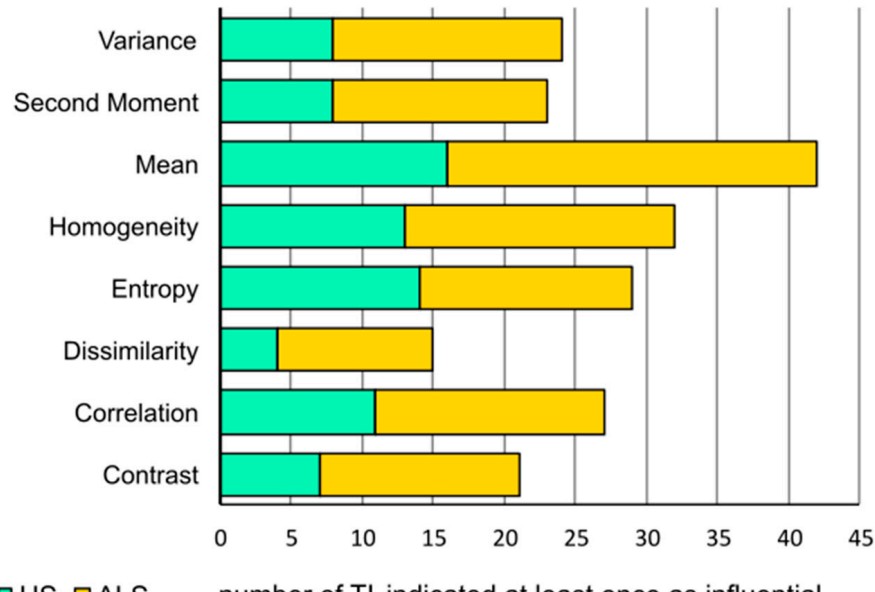

**Figure 5.** The number of influential TI acquired for HS and ALS. The figure shows how many times each type of TI was indicated at least once as influential (the PI value was above 0) divided into HS and ALS data.

For experiment 2, it was decided to analyse features that were indicated as influential five times or more out of 10 PI iterations. The HS and ALS data were analysed separately. The mean layer calculated for 14 different ALS or HS layers was influential: five from ALSF, five from TOPO and four from MNF. Moreover, only one entropy layer, calculated based on MNF, was found to be influential in five PI iterations. As a result, in experiment 2, the mean and entropy were calculated for HS data and the mean for ALS data.

In the case of ALS data, only mean TI was calculated for all TOPO and chosen ALSF. In the case of ALSF, most of the layers were not influential in the classification based on PI results. To reduce the amount of data, only 24 out of 93 ALSF (those ALSF for which calculated TI were indicated as influential at least one time) were chosen to calculate mean TI. From nine TOPO bands, nine mean TI were calculated. Experiment 2 resulted in four scenarios: HS, HS + TI$_{HS}$, HS + ALS and HS + ALS + TI$_{HS+ALS}$ (Tables 3 and A4).

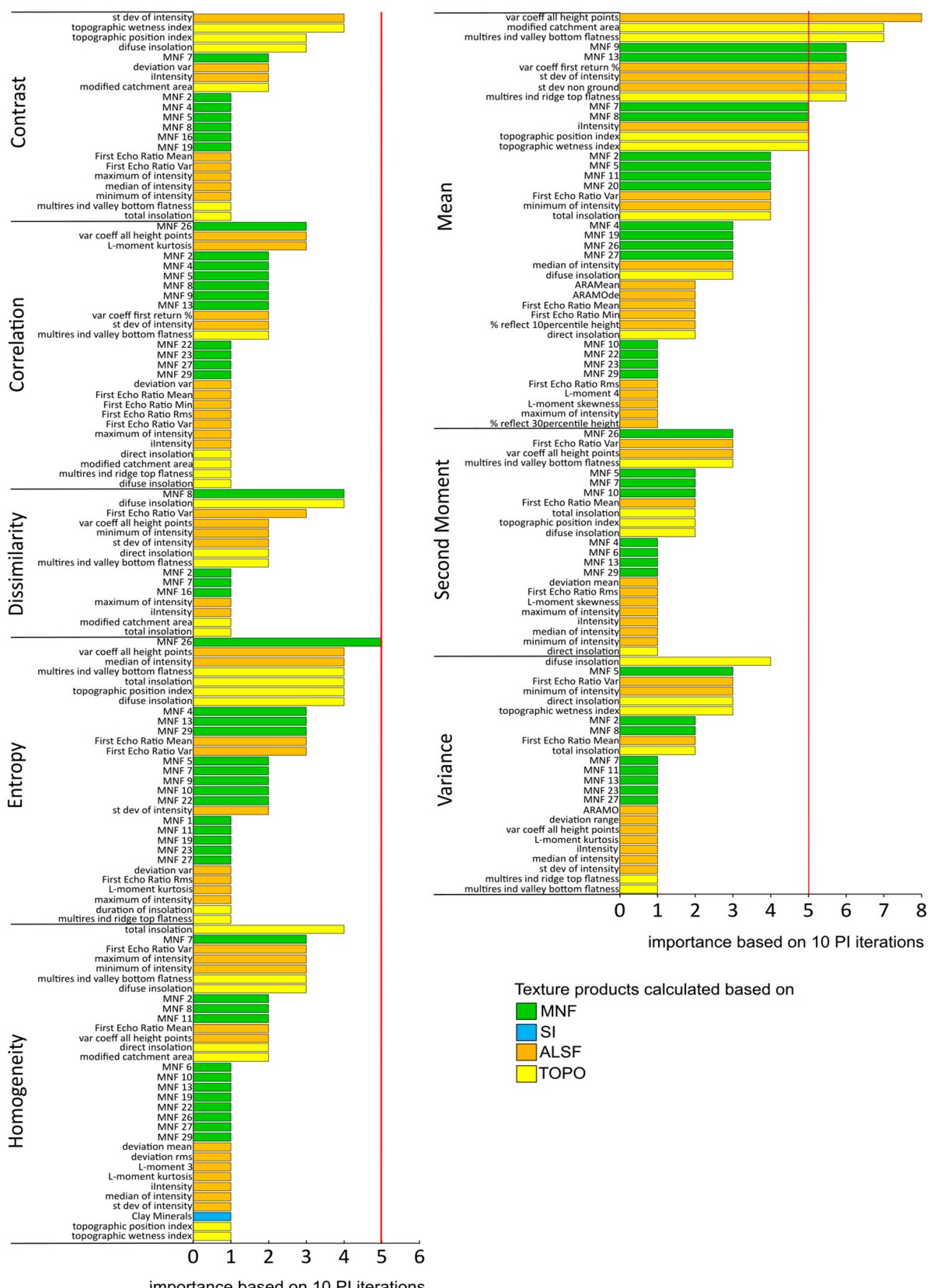

**Figure 6.** Information on how many times out of 10 iterations a TI feature was indicated as influential by the PI algorithm. Usability is defined by the PI algorithm as a mean decrease in the F1 macro score (after a random shuffle of the feature value). For values above 0, the layer was indicated as influential in the iteration. On the figure are presented only layers that were indicated as influential at least once.

**Table 3.** The four scenarios used for the classification of the whole study area (Experiment 2).

| Nb | Scenario | Products | Layers |
|---|---|---|---|
| 1 | HS | HS data: MNF, SI | 37 |
| 2 | HS + TI$_{HS}$ | HS data: MNF, SI, TI (mean and entropy) calculated based on HS data | 111 |
| 3 | HS + ALS | HS and ALS data: MNF, SI, ALSF and TOPO | 139 |
| 4 | HS + ALS + TI$_{HS+ALS}$ | HS and ALS data: MNF, SI, TI (mean and entropy) calculated based on HS data; ALSF and TOPO, and mean TI calculated based on ALS data | 246 |

*4.2. The Accuracy of Wetland Communities Mapping Depending on the Use of TI (Experiment 2)*

4.2.1. The Influence of TI on Classification Accuracy Using HS Data

The result of the second experiment was classifications performed for the entire study area (Figure 7, Table 4).

**Table 4.** The average F1 values calculated based on 50 iterations of four scenarios.

| F1 Value (Mean) | HS | HS + TI$_{HS}$ | HS + ALS | HS + ALS + TI$_{HS+ALS}$ |
|---|---|---|---|---|
| average F1 | 0.730 * | 0.735 * | 0.764 * | 0.775 * |
| *Lemnetea* and *Potametea* | 0.800 | 0.799 | 0.834 | 0.837 |
| *Phalaridetum arundinaceae* | 0.681 | 0.684 | 0.707 * | 0.735 * |
| *Magnocaricion* | 0.793 | 0.796 | 0.784 | 0.791 |
| *Phragmition* | 0.681 | 0.688 | 0.724 * | 0.740 * |
| *Isoëto-Nanojuncetea* | 0.346 | 0.353 | 0.381 | 0.393 |
| *Bidentetea* | 0.650 | 0.643 | 0.643 | 0.643 |
| *Trifolio-Agrostietalia* and *Plantaginetalia* | 0.665 | 0.671 | 0.777 | 0.780 |
| *Molinietalia* | 0.674 | 0.676 | 0.705 | 0.710 |
| *Arrhenatheretalia* | 0.462 | 0.436 | 0.459 | 0.446 |
| *Koelerio-Corynephoretea* and *Festuco-Brometea* | 0.787 | 0.786 | 0.803 | 0.812 |
| *Artemisietea* and *Epilobietea* | 0.510 * | 0.523 * | 0.618 | 0.632 |
| *Salicetea purpureae* | 0.867 * | 0.904 * | 0.894 * | 0.925 * |
| *Ribeso nigri-Alnetum* and *Alno-Ulmion* | 0.928 | 0.928 | 0.943 * | 0.960 * |
| *Salicetum pentandro-cinereae* | 0.843 * | 0.913 * | 0.885 * | 0.925 * |
| *Vaccinio-Piceetea* | 0.885 | 0.875 | 0.854 | 0.860 |
| Others wooded communities | 0.684 | 0.684 | 0.820 * | 0.836 * |
| Areas without vegetation | 0.936 * | 0.921 * | 0.963 | 0.966 |
| Surface water | 0.944 | 0.944 | 0.960 | 0.959 |

* The differences in the average are statistically significant.

The mean and distribution of F1 values for each of the 18 classes were used to assess differences between the HS and HS + TI$_{HS}$ scenarios. Based on the average accuracy of F1 for all classes, it can be concluded that the results for the HS scenario (mean F1 for 18 classes was equal to 0.730) are lower than those of HS + TI$_{HS}$ (F1 = 0.735), and the difference is statistically significant based on the *t*-test.

For the HS dataset, the average accuracy of F1 for a single class varied from 0.346 (*Isoëto-Nanojuncetea*), which is a very low value and the class was not properly classified, to 0.928 (*Ribeso nigri-Alnetum* and *Alno-Ulmion*), which indicate correct identification. For seven classes, including five communities classes (*Lemnetea* and *Potametea, Salicetea purpureae, Ribeso nigri-Alnetum* and *Alno-Ulmion, Salicetum pentandro-cinereae, Vaccinio-Piceetea*), the F1 accuracy was above 0.800, which indicates the correct classification of these units. An average F1 value was below 0.500 for two classes only (*Isoëto-Nanojuncetea, Arrhenatheretalia*). Similar conclusions can be drawn from the results calculated based on the HS + TI$_{HS}$ scenario. For the same two classes (*Isoëto-Nanojuncetea, Arrhenatheretalia*), the accuracies were below 0.500; and for four classes, the F1 were above 0.800 (*Salicetea purpureae, Ribeso nigri-Alnetum* and *Alno-Ulmion, Salicetum pentandro-cinereae, Vaccinio-Piceetea*).

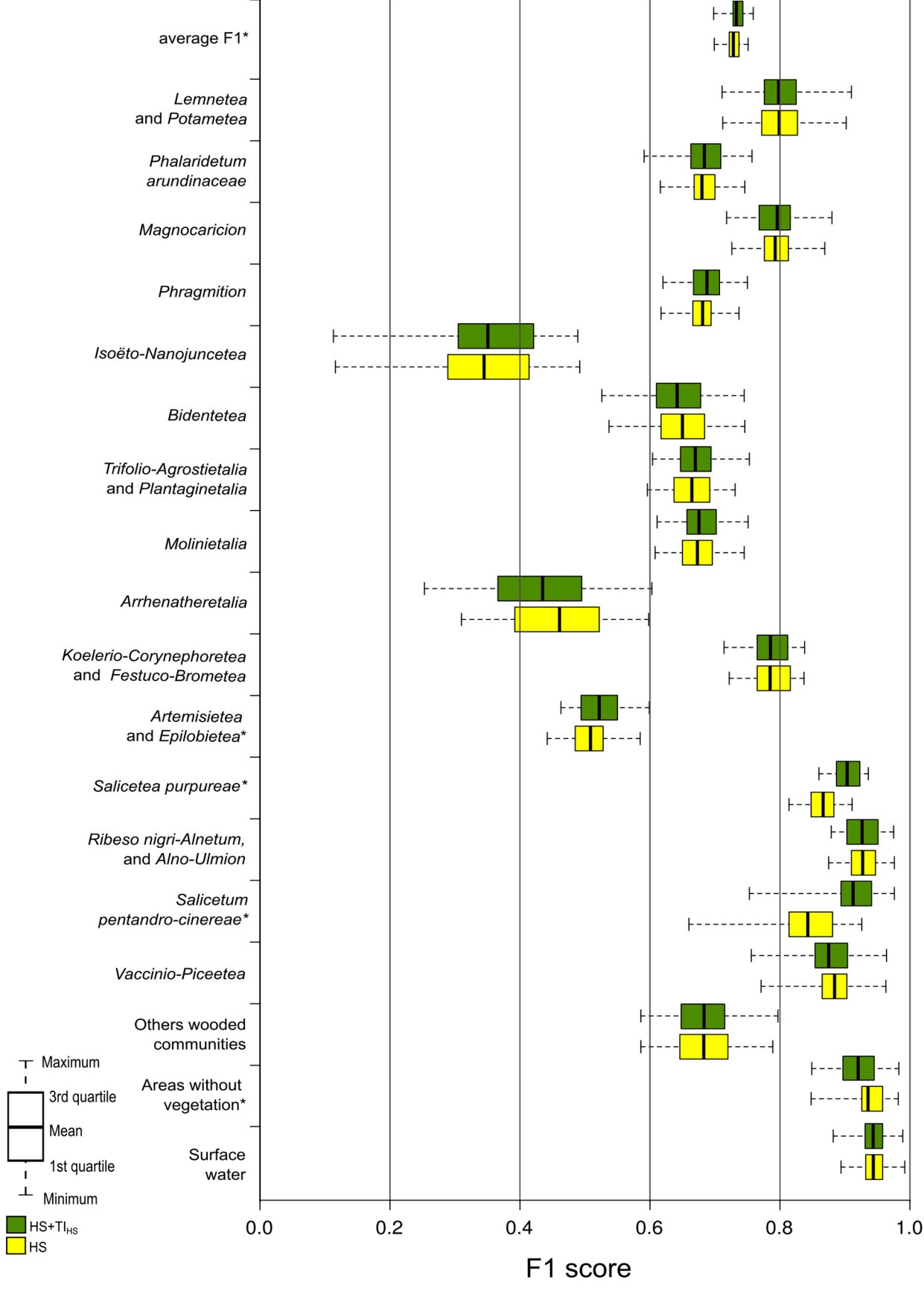

**Figure 7.** The distribution of F1 values calculated in 50 iterations of the CatBoost algorithm for HS and HS + TI$_{HS}$ scenarios. Differences for classes marked with * were statistically significant.

Based on the accuracies acquired for the two scenarios, HS and HS + TI$_{HS}$, it can be concluded that the differences are not high. For nine classes, adding TI to the HS data increased the F1 accuracy: *Phalaridetum arundinaceae, Magnocaricion, Phragmition, Isoëto-Nanojuncetea, Trifolio-Agrostietalia* and *Plantaginetalia, Molinietalia, Artemisietea* and *Epilobietea, Salicetea purpureae* and *Salicetum pentandro-cinereae*, but the differences were statistically not significant and do not exceed 0.007. The F1 values for other wooded communities and surface water were almost the same for both scenarios.

In the case of four classes, the differences in average F1 were statistically significant: *Artemisietea* and *Epilobietea, Salicetea purpureae, Salicetum pentandro-cinereae* and areas without vegetation. The largest differences (0.070) were noted for the *Salicetum pentandro-cinereae*, and slightly lower (0.036) for *Salicetea purpureae*. For only one class—areas without vegetation—the F1 value was higher (by 0.015) for the HS scenario compared to the HS + TI$_{HS}$.

For the forests and bushes classes (*Salicetea purpureae, Ribeso nigri-Alnetum* and *Alno-Ulmion, Salicetum pentandro-cinereae, Vaccinio-Piceetea* and other wooded communities), the accuracy is improved by adding TI—the average F1 for the HS scenario was equal to 0.841 and for HS + TI$_{HS}$, 0.861. Moreover, better results for scenarios with TI were noted for rushes (*Phalaridetum arundinaceae, Magnocaricion, Phragmition*): 0.718 for HS and 0.723 for HS + TI$_{HS}$.

### 4.2.2. The Influence of TI on Classification Accuracy Using HS and ALS Data

The results of the second comparison of the two scenarios, HS + ALS and HS + ALS + TI$_{HS+ALS}$, generally indicate that the accuracy for the scenario with TI is slightly higher (Figure 8, Table 4). The mean value of F1 for all 18 classes is higher for HS + ALS + TI$_{HS+ALS}$ (F1 = 0.775) compared to the HS + ALS scenario (F1 = 0.764), and this small difference is statistically significant based on the *t*-test.

For the HS + ALS scenario, the average F1 for a single class varied from *Isoëto-Nanojuncetea* (0.381), which indicates poor quality of identification, to 0.943 for *Ribeso nigri-Alnetum* and *Alno-Ulmion*, and even higher for two non-vegetation classes: areas without vegetation (0.963) and surface water (0.960). For nine classes, including communities classes (*Lemnetea* and *Potametea, Koelerio-Corynephoretea* and *Festuco-Brometea, Salicetea purpureae, Ribeso nigri-Alnetum* and *Alno-Ulmion, Salicetum pentandro-cinereae, Vaccinio-Piceetea* and other wooded communities) the F1 values exceed 0.800, so these areas are properly classified. Two classes (*Isoëto-Nanojuncetea, Arrhenatheretalia*) were not correctly classified, with the average class accuracy below 0.500.

The general results for the HS + ALS + TI$_{HS+ALS}$ scenario are similar to HS + ALS—the F1 values varied from 0.393 for *Isoëto-Nanojuncetea* to 0.960 for *Ribeso nigri-Alnetum* and *Alno-Ulmion* and higher for areas without vegetation (0.966). The same two classes as for HS + ALS were poorly classified (F1 less than 0.500): *Isoëto-Nanojuncetea* and *Arrhenatheretalia*. An average F1 higher than 0.800 and good classification quality were noted for the same classes as for HS + ALS: *Lemnetea* and *Potametea, Koelerio-Corynephoretea* and *Festuco-Brometea, Salicetea purpureae, Ribeso nigri-Alnetum* and *Alno-Ulmion, Salicetum pentandro-cinereae, Vaccinio-Piceetea*, other wooded communities, areas without vegetation, and surface water.

For almost all of the classes, the F1 value was higher for the HS + ALS + TI$_{HS+ALS}$ scenario than for HS + ALS, although the differences were not high. The highest differences were noted for *Salicetum pentandro-cinereae* (+0.040), *Salicetea purpureae* (+0.030) and *Phalaridetum arundinaceae* (+0.027). Only for the *Arrhenatheretalia* class was the F1 accuracy for HS + ALS higher (0.446) compared to HS + ALS + TI$_{HS+ALS}$ (0.459). No difference in F1 between the two scenarios was noted for *Bidentetea* and surface water. For six classes (*Phalaridetum arundinaceae, Phragmition, Salicetea purpureae, Ribeso nigri-Alnetum* and *Alno-Ulmion, Salicetum pentandro-cinereae* and other wooded communities), the differences were statistically significant and average accuracy was higher for the HS + ALS + TI$_{HS+ALS}$

compared to the HS + ALS scenario for these classes. Within these five classes, the smallest difference was noted for *Phragmition* (0.015).

For the forests and bushes classes, the accuracy improved by adding TI, and the differences were higher compared to previous scenarios: for HS + ALS and HS + ALS + TI$_{HS+ALS}$, the F1 values were equal to 0.879 and 0.901, respectively. The same situation was noted for rushes—the average F1 value for HS + ALS + TI$_{HS+ALS}$ was higher by 0.017 than for the HS + ALS scenario.

### 4.2.3. Influence of TI on the Effectiveness of Patches Delineation Based on Acquired Communities Maps

A comparative analysis of the prediction maps showed clear differences between the two types of scenarios (without TI − HS and HS + ALS and with TI − HS + TI$_{HS}$ and HS + ALS + TI$_{HS+ALS}$) (Figure 9). The main difference was the weaker "salt and pepper" effect using TI. In addition, the patches of each class were less fragmented, and the boundaries between patches were clearer and easier to distinguish. As a result of these differences, maps with the application of TI better represented the spatial differentiation of plant communities occurring in the analysis area. Adding TI to both scenarios (HS and HS + ALS) resulted in a clear reduction in the visibility of the mosaic line, which resulted in an improvement in the course of the boundary between patches.

To compare the results for single classes, four areas of interest were analysed A, B, C, and D (see Figure 9), and the differences between scenarios were observed. For area A in the HS scenario, *Phalaridetum arundinaceae* patches were weakly isolated. Adding TI to both scenarios (HS and HS + ALS) resulted in an improvement in the course of the boundary between patches: for HS + TI$_{HS,}$ the improvement was noted for *Phalaridetum arundinaceae*, whereas for HS + ALS + TI$_{HS+ALS}$, the differences were also visible for the *Molinietalia*, *Trifolio-Agrostietalia* and *Plantaginetalia*, and *Artemisietea* and *Epilobietea* classes.

In area B, in the HS and HS + ALS scenarios, patches of annual *Bidentetea* were classified in a shallow depression, and areas of *Phragmition* and *Artemisietea* and *Epilobietea* were highly fragmented and surrounded by patches of *Trifolio-Agrostietalia* and *Plantaginetalia* pastures. Adding TI to the classifications resulted in clearly separating patches of *Bidentetea* and *Artemisietea,* and *Epilobietea* from patches of *Trifolio-Agrostietalia* and *Plantaginetalia*, which is consistent with the actual co-occurrence of patches of these plant communities.

The C area maps produced with the HS and HS + ALS scenarios were dominated by medium-sized patches of *Artemisietea* and *Epilobietea* and highly fragmented patches of *Phragmition*. Using the HS + TI$_{HS}$ and HS + ALS + TI$_{HS+ALS}$ scenarios resulted in a decrease in the *Artemisietea* and *Epilobietea* area, in favour of rush classes (*Phalaridetum arundinaceae* and *Phragmition*), which better correspond to the ground situation.

For area D, in both scenarios without TI, a very strong fragmentation of the patches of each class was visible. Moreover, not very clear boundaries between patches were noted: for the HS scenario in *Molinietalia*, *Phalaridetum arundinaceae* and *Trifolio-Agrostietalia* and *Plantaginetalia* classes; for the HS + ALS scenario in *Molinietalia* and *Trifolio-Agrostietalia* and *Plantaginetalia*. On the maps produced by the HS + TI$_{HS}$ and HS + ALS + TI$_{HS+ALS}$ scenarios, the course of the boundary between patches of the mentioned classes was improved.

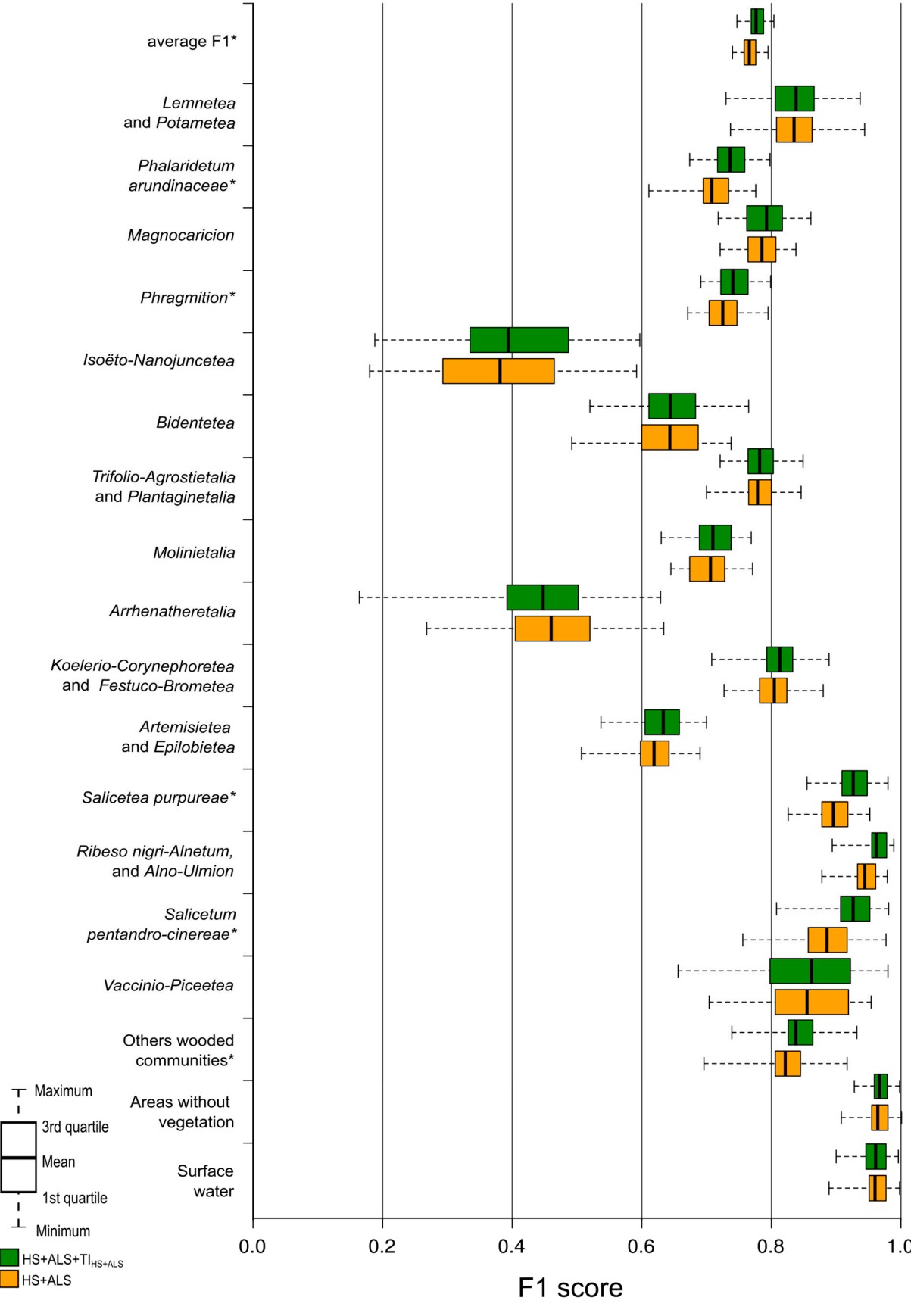

**Figure 8.** The distribution of F1 values calculated in 50 iterations of CatBoost algorithm for HS and HS + ALS + TI$_{HS+ALS}$ scenarios. Differences for classes marked with * were statistically significant.

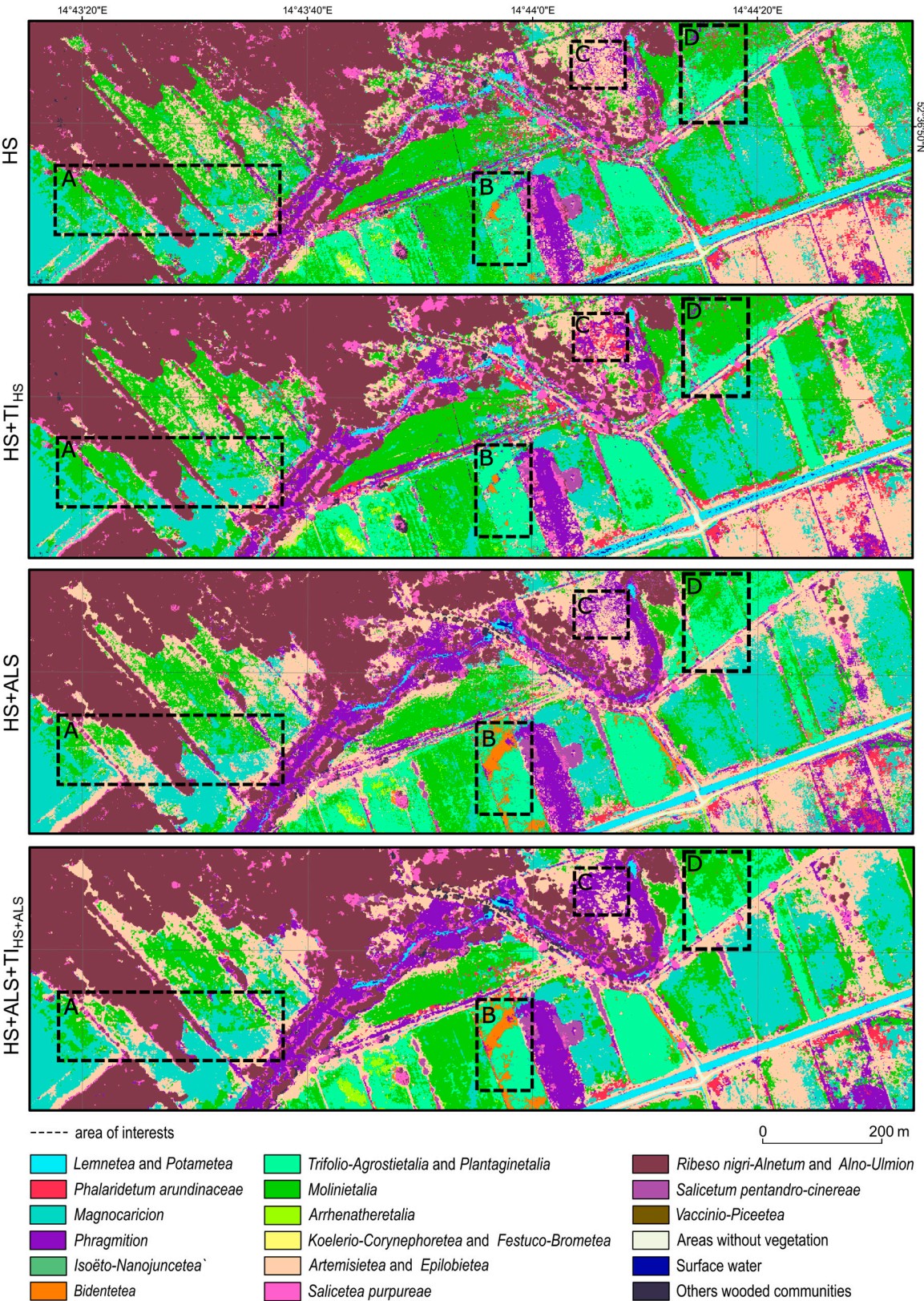

**Figure 9.** The communities map calculated based on 50 CatBoost classifications for four scenarios performed on a part of the Park. The area is marked in Figure 1. Areas marked with A, B, C and D are described in detail in the text.

## 5. Discussion

### 5.1. The Utility of TI in Wetland Communities Mapping

The research results indicate that the mean TI, calculated based on HS and ALS data, and entropy TI, calculated using HS images, are useful for differentiation (Figures 5 and 6). Similar information was acquired in other studies. In the case of bamboo forest classification, mean TI was defined as the most useful [52]. For tree species identification, the mean was also one of the important TI, together with contract and dissimilarity, or mean and correlation, dependent on the analysis [53]. In other studies, in rice leaf blast classification, one useful TI, among others, was entropy [54]. To sum up, mean and entropy were also useful in other studies.

Adding TI to the dataset, whether HS only or HS and ALS, improved the classification results for most of the classes by about 0.01–0.02 based on the F1 value (Figures 7 and 8). Such results can also be found in other studies, mainly concerning trees. For forest species classification based on the multispectral images, overall accuracy improved by about 0.12 for one study area and 0.05 for a second one by adding textural information [55]. For the classification of Australian woodlands, the OA was improved by 0.05 by adding TI to the hyperspectral HyMap images [19]. The differences for forests classification are quite clear, whereas the differences in this study for F1 values are not high (the maximum difference was 0.07 for *Salicetum pentandro-cinereae* between HS and HS + $TI_{HS}$) but visible on acquired prediction maps (Figure 9). The patch borders are more clearly recognized, and the "salt and pepper" effect is less intensive compared to the maps without TI used as input layers.

The differences are strongly dependent on the analysed class. For the forests and bushes classes, the accuracy is improved by adding TI: for these classes, *Salicetea purpureae*, *Ribeso nigri-Alnetum* and *Alno-Ulmion*, *Salicetum pentandro-cinereae*, *Vaccinio-Piceetea* and other wooded communities, the F1 value was higher. The average F1 for these communities for the HS scenario was equal to 0.841 and for HS + $TI_{HS}$, 0.861. The differences were even higher for the HS + ALS and HS + ALS + $TI_{HS+ALS}$ scenarios—the F1 values were equal to 0.879 and 0.901, respectively. Probably the better accuracy is related to the size of individual elements of the habitat. A single tree or bush is big enough to be highlighted by the TI layers. In the case of rushes (*Phalaridetum arundinaceae*, *Magnocaricion* and *Phragmition*), the use of TI also increased the F1 accuracy. These are communities with one dominant species, similar in structure to arable fields. In this case, the use of TI mean, and entropy emphasizes the internal homogeneity of the community. Similar conclusions can be drawn for arable land [54]. The TI did not significantly change the accuracy within classes that were very heterogeneous, where it was not possible to distinguish individual elements due to the limited spatial resolution of the data. Moreover, individual objects (plants) in these communities are smaller than the GSD size—smaller than 1 m$^2$.

There is a less visible "salt and pepper" effect on maps classified with TI. Moreover, the borders of flight lines are less visible. This is probably due to the use of the TI mean, which averages the pixel values. Additionally, on maps classified using TI, patch borders are better defined. This is probably related to the better identification of individual objects within classes. This difference in the map is not reflected in F1 accuracy because the reference polygons were located inside the communities patches, not close to the border. In order to be able to assess the accuracy of the entire map and the patch borders, it is necessary to collect reference data at the borders.

### 5.2. Applicability of the Results

The conducted analysis was performed to identify the possibility of better vegetation identification. In natural ecosystems, such as wetlands, the vegetation is diverse and correct delineation can be difficult. Based on the results, it can be concluded that TI mean and entropy are useful for the classification of forest and shrub areas as well as communities with one dominant species. The differences in accuracy metrics are not high, but the use of TI improves the delineation of patches. The better identification of patch boundaries is the basis for maps to be useful in environmental protection management. So, in the case of

forests, shrubs and communities with one dominant species, it is recommended to add TI to the dataset.

On the other hand, in the case of grassland and meadow communities, TIs with the applied spatial resolution will probably not increase accuracy. In this case, it was not possible to capture the high internal differentiation of the vegetation in TI.

In future studies, it is necessary to test the TI calculated using data with a higher than HS data spatial resolution—for example, an orthophoto map. In this case, it would be possible to recognize individual objects of meadow and grassland communities.

Based on the analyses, it can be concluded that the method of including TI mean, and entropy in order to increase the F1 accuracy of the classification can be implemented without prior Permutation Importance analysis. However, the study area has to be geographically comparable with similar community classes. In addition, it is also necessary to use comparable input data.

## 6. Conclusions

Research conducted in the Warta River Mouth National Park (Poland) using HS and ALS data indicates that:

1. The textural information with the highest information potential for identifying wetland communities are mean for HS and ALS data and entropy for HS data;
2. The addition of textural information in the dataset leads to an increase in mean F1 accuracy of 0.005 when using HS data and 0.011 when using a fusion of HS and ALS data;
3. The resulting maps from the scenarios using TI allow for better delineation of the patch boundaries of individual community units and eliminate the "salt and pepper" effect and the visibility of mosaic lines. In order to analyse this change in quality in the maps, it is necessary to have verification polygons located as close as possible to the real patch boundary. Since there was a small proportion of polygons close to the patch boundary in the reference dataset, the changes in accuracy measures were also smaller than the visual differences;
4. A comparison of the classification with TI and without TI shows the greatest increase in accuracy after the application of TI for scrub and forest communities (by 0.019 for the scenarios with HS and 0.022 for the scenarios with HS + ALS).

In summary, it can be concluded that the use of textural information allows for more precise mapping of wetland communities. Further research is needed to test textural information derived from other data types, such as high-resolution RGB orthophotos.

**Author Contributions:** Conceptualization, A.J., D.K. and J.N.; methodology, A.J.; software, J.C.; validation, D.K.; formal analysis, A.J., J.N. and J.C.; investigation, A.J., J.W., B.O. and D.K.; resources, J.N.; data curation, J.N.; writing—original draft preparation, A.J., D.K., J.W., J.N., B.O. and J.C.; writing—review and editing, A.J. and D.K.; visualization, A.J. and B.O.; supervision, D.K.; project administration, D.K.; funding acquisition, D.K. All authors have read and agreed to the published version of the manuscript.

**Funding:** Remote sensing and reference data were made as part of the project entitled: "Assessment of the state of natural resources of the "Ujście Warty" National Park and valuable fragments of its buffer zone using modern remote sensing techniques combined with the development of the interoperable Park Spatial Information System". The project is financed by the European Union—Operational Programme Infrastructure and Environment 2014–2020 under the program 2.4.4d—assessment of the state of natural resources in national parks using modern remote sensing techniques, competition call POIS.02.04.00-IW.02-004D1/17. The publication was co-financed by the University of Warsaw.

**Data Availability Statement:** The airborne images were acquired by the MGGP Aero company and delivered to the National Park "Ujście Warty", which is the owner of the data. Reference polygons were acquired during field mapping.

**Acknowledgments:** The authors are grateful to the National Park "Ujście Warty" for the possibility of using airborne remote sensing data and permission to conduct field research in the park.

**Conflicts of Interest:** The authors declare no conflict of interest.

## Appendix A

**Table A1.** Spectral indices calculated based on hyperspectral bands.

| Index | Formula | Source |
|---|---|---|
| Anthocyanin Reflectance Index 2 | ARI2 = R800(1/R550 − 1/R700) | [56] |
| Carotenoid Reflectance Index 1 | CRI1 = 1/R510 − 1/R550 | [57] |
| Clay Minerals | CM = R1650/R2215 | [58] |
| Iron Oxide | IO =R660/R485 | [59] |
| Normalized Difference Nitrogen Index | NDNI = (log(1/R1510) − log(1/R1680))/(log(1/R1510) + log(1/R1680)) | [60] |
| Red Green Ratio Index | RGRI = ($\sum$(i = 600)^699 Ri)/($\sum$(j = 500)^599 Rj) | [61] |
| WorldView Water Index | WWI = (R427 − R950)/(R427 + R950) | [62] |

**Table A2.** The list of ALS features (ALSF).

| ALSF | Description | Source |
|---|---|---|
| ARAMean | All returns above mean divided by (total first returns) × 100 | [63] |
| ARAMOde | All returns above mode divided by (total first returns) × 100 | [63] |
| 1st decile of height | 10th percentile of height values | [41] |
| 2nd decile of height | 20th percentile of height values | [41] |
| 3nd decile of height | 30th percentile of height values | [41] |
| 4nd decile of height | 40th percentile of height values | [41] |
| 5nd decile of height | 50th percentile of height values | [41] |
| 6nd decile of height | 60th percentile of height values | [41] |
| 7nd decile of height | 70th percentile of height values | [41] |
| 8nd decile of height | 80th percentile of height values | [41] |
| 9nd decile of height | 90th percentile of height values | [41] |
| Deviation max | Maximum value of deviation from pulse shape in the grid cell | [41] |
| Deviation mean | Mean value of deviation from pulse shape in the grid cell | [41] |
| Deviation median | Median value of deviation from pulse shape in the grid cell | [41] |
| Deviation min | Minimum value of deviation from pulse shape in the grid cell | [41] |
| Deviation range | Range of deviation values from pulse shape in the grid cell | [41] |
| Deviation rms | Squared mean value of deviation from pulse shape in the grid cell | [41] |
| Deviation var | Variance of deviation from pulse shape in the grid cell | [41] |
| 25th percentile of dev | 25th percentile of deviation from pulse shape in the grid cell | [41] |
| 75th percentile of dev | 75th percentile of deviation from pulse shape in the grid cell | [41] |
| Largest eigen of the cov matrix | Largest eigenvalue of the covariance matrix of the points 3D position in the grid cell | [41] |
| Medium eigen of the cov matrix | Largest eigenvalue of the covariance matrix of the points 3D position in the grid cell | [41] |
| Smallest eigen of the cov matrix | Largest eigenvalue of the covariance matrix of the points 3D position in the grid cell | [41] |
| Fraction of first return | Fraction of first return pulses intercepted by tree | [64] |
| First_Echo_Ratio_Mean | Mean value of number of points defined in 3D fixed neighborhood divided by number of points defined in fixed 2D neighborhood in the grid cell | [65] |

**Table A2.** *Cont.*

| ALSF | Description | Source |
|---|---|---|
| First_Echo_Ratio_Min | Min value of number of points defined in 3D fixed neighborhood divided by number of points defined in fixed 2D neighborhood in the grid cell | [65] |
| First_Echo_Ratio_Range | Range of values of number of points defined in 3D fixed neighborhood divided by number of points defined in fixed 2D neighborhood in the grid cell | [65] |
| First_Echo_Ratio_Rms | Root mean square of values of number of points defined in 3D fixed neighborhood divided by number of points defined in fixed 2D neighborhood in the grid cell | [65] |
| First_Echo_Ratio_Var | Variance of values of number of points defined in 3D fixed neighborhood divided by number of points defined in fixed 2D neighborhood in the grid cell | [65] |
| Fraction of all returns | Fraction of all returns classified as tree | [64] |
| Max height above gr first returns | Maximum height above ground of all first returns | [66] |
| 90th–25th perc | 90th percentile–25th percentile of height values | [67] |
| 90th–50th perc | 90th percentile–50th percentile of height values | [67] |
| 99th–25th perc | 99th percentile–25th percentile of height values | [67] |
| 99th–50th perc | 99th percentile–50th percentile of height values | [67] |
| Var coeff all height points | The coefficient of variation of all height points within each pixel | [68] |
| Var coeff first return % | Coefficient of variation percentage of heights of all first returns relative to all returns | [66] |
| L-moment 1 | 1st L-moment of height values | [63] |
| L-moment 2 | 2st L-moment of height values | [63] |
| L-moment 3 | 3st L-moment of height values | [63] |
| L-moment 4 | 4st L-moment of height values | [63] |
| L-moment kurtosis | L-moment kurtosis of height values | [63] |
| L-moment skewness | L-moment skewness of height values | [63] |
| MADev from Median Height | The Median Absolute Deviation from Median Height value (HMAD) of all height points within each pixel, where HMAD = $1.4826 \times$ median ($\lvert$height − median height$\rvert$) | [68] |
| MADev from overall mode | Median of the absolute deviations from the overall mode | [63] |
| Horizontality | Measure of horizontality of points based on eigenvalues of the covariance matrix of the points 3D position in the grid cell | [41] |
| 25th Percentile intensity | 25th Percentile of intensity | [67] |
| 50th Percentile intensity | 50th Percentile of intensity | [67] |
| 75th Percentile intensity | 75th Percentile of intensity | [67] |
| 99th Percentile intensity | 99th Percentile of intensity | [67] |
| Kurtosis of intensity | Kurtosis of Intensity | [69] |
| Kurtosis of reflectance | Kurtosis of Reflectance | [69] |
| Maximum of intensity | Maximum of Intensity | [69] |
| Maximum of reflectance | Maximum of Reflectance | [69] |
| Mean of intensity | Mean of Intensity | [69] |
| Mean of reflectance | Mean of Reflectance | [69] |
| Median of intensity | Median of Intensity | [69] |
| Median of reflectance | Median of Reflectance | [69] |
| Minimum of intensity | Minimum of Intensity | [69] |
| Minimum of reflectance | Minimum of Reflectance | [69] |
| % intens 10percentile height | Percentage of intensity values for heights below the 10th percentile of heights | [41] |
| % reflect 10percentile height | Percentage of reflectance values for heights below the 10th percentile of heights | [41] |

**Table A2.** *Cont.*

| ALSF | Description | Source |
|---|---|---|
| % intens 30percentile height | Percentage of intensity values for heights below the 30th percentile of heights | [41] |
| % reflect 30percentile height | Percentage of reflectance values for heights below the 30th percentile of heights | [41] |
| % intens 50percentile height | Percentage of intensity values for heights below the 50th percentile of heights | [41] |
| % reflect 50percentile height | Percentage of reflectance values for heights below the 50th percentile of heights | [41] |
| % intens 70percentile height | Percentage of intensity values for heights below the 70th percentile of heights | [41] |
| % reflect 70percentile height | Percentage of reflectance values for heights below the 70th percentile of heights | [41] |
| % intens 90percentile height | Percentage of intensity values for heights below the 90th percentile of heights | [41] |
| % reflect 90percentile height | Percentage of reflectance values for heights below the 90th percentile of heights | [41] |
| Interquartile range of dev | Interquartile range (P75–P25) of deviation from pulse shape in the grid cell | [41] |
| Interquartile range of dev | Interquartile range (P75–P25) of deviation from pulse shape in the grid cell | [41] |
| Range of reflectance | Range of reflectance | [67] |
| Values | values | [67] |
| St dev of intensity | Standard deviation of intensity | [69] |
| St dev of reflectance | Standard deviation of reflectance | [69] |
| Skewn of intensity | Skewness of intensity | [69] |
| Skewn of reflectance | Skewness of reflectance | [69] |
| Linearity | Measure of linearity of points based on eigenvalues of the covariance matrix of the points 3D position in the grid cell | [41] |
| Median abs dev | Median absolute deviation = median ($|$height − median height$|$) of tree returns Meters MAD | [64] |
| Nb of points below GT | The total number of all the points within each pixel that are below the specified Ground Threshold value (GT) | [68] |
| Nb of modes | Number of Modes | [70] |
| St dev non ground | Standard deviation of heights for points between 0 and 1 m | [41] |
| Nb of points above CT | The total number of all the points within each pixel that are above the specified Crown Threshold value (CT) | [68] |
| % returns above mean | Percentage all returns above mean/total all returns | [63] |

**Table A3.** The list of Topographic indices list (TOPO).

| Feature | Index Full Name |
|---|---|
| direct insolation | Direct Insolation |
| duration of insolation | Duration of Insolation |
| modified catchment area | Modified Catchment Area |
| multi-resolution ridge top flatness | Multi-resolution index of the Ridge Top Flatness |
| multi-resolution valley bottom flatness | Multi-resolution Index of Valley Bottom Flatness |
| total insolation | Total Insolation |
| topographic position index | Topographic Position Index |
| topographic wetness index | Topographic Wetness Index |
| diffuse insolation | Diffuse Insolation |

**Table A4.** Textural Feature formulas, where P(i)—the probability of each pixel value and the variable μ represents the mean of P.

| Feature | Formula |
|---------|---------|
| Mean | $\sum_{i=1}^{N_g} \sum_{j=1}^{N_g} i \times P(i, j)$ |
| Variance | $\sum_{i=1}^{N_g} \sum_{j=1}^{N_g} (i - \mu)^2 \times P(i, j)$ |
| Homogeneity | $\sum_{i=1}^{N_g} \sum_{j=1}^{N_g} \frac{1}{1+(i-j)^2} \times P(i, j)$ |
| Contrast | $\sum_{i=1}^{N_g} \sum_{j=1}^{N_g} P(i, j)(i - j)^2$ |
| Dissimilarity | $\sum_{i=1}^{N_g} \sum_{j=1}^{N_g} P(i, j)|i - j|$ |
| Entropy | $\sum_{i=1}^{N_g} \sum_{j=1}^{N_g} P(i, j) \log(P(i, j))$ |
| Second Moment | $\sum_{i=1}^{N_g} \sum_{j=1}^{N_g} \{P(i, j)\}^2$ |
| Correlation | $\frac{\sum_i \sum_j (i, j) P(i, j) - \mu_x \mu_y}{\sigma_x \sigma_y}$ |

**Table A5.** The four scenarios layers used for classification in experiment 2.

| Nb | Scenario | Products |
|----|----------|----------|
| 1 | HS | HS data: 30 MNF, 7 SI |
| 2 | HS + TI$_{HS}$ | HS data: 30 MNF, 7 SI, texture features (mean and entropy) calculated based on HS bands |
| 3 | HS + ALS | HS and ALS data: 30 MNF, 7 SI, 93 ALSF and 9 TOPO |
| 4 | HS + ALS + TI$_{HS+ALS}$ | HS and ALS data: 30 MNF, 7 SI, mean and entropy calculated based on HS data; 93 ALSF, 9 TOPO, 24 mean texture features calculated based on chosen ALSF(ARAMean, ARAMOde, deviation mean, deviation range, deviation rms, deviation var, duration of insolation, First_Echo_Ratio_Mean, First_Echo_Ratio_Min, First_Echo_Ratio_Rms, First_Echo_Ratio_Var, var coeff all height points, var coeff first return %, L-moment 3, L-moment 4, L-moment kurtosis, L-moment skewness, maximum of intensity, mean of intensity, median of intensity, % reflect 10percentile height, % reflect 30percentile height, st dev of intensity, st dev non ground) and 9 mean texture features calculated using TOPO products |

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
