# Peer review of "Testing Textural Information Base on LiDAR and Hyperspectral Data for Mapping Wetland Vegetation: A Case Study of Warta River Mouth National Park (Poland)"

_remotesensing, doi:10.3390/rs15123055_

Round 1
Reviewer 1 Report
Dear authors,
The research is well-designed and presented, therefore I would recommend publications without further comments.
Congratulations,
Author Response
We are much obliged to you for your time and positive feedback.
Reviewer 2 Report
The research is useful and important, describing the use of remotely sensed textural information to improve classification of wetland communities. The understanding, use of literature and information provided are all very good. Research questions are clear and concise. The methods appear to be very thorough and well described. The results section and figures are useful and informative. The discussion is very thorough and provides useful conclusions. However, the language needs checking throughout, there are many grammatical errors. The language of the paper is very technical with many acronyms, for this paper to have wider appeal beyond technical specialists it would be desirable to have a bit more explanation of terms used (can be in supplementary) and some simplification in key sections e.g. discussion.
Abstract
It would be helpful if the abstract contained an extra sentence explaining how remote sensing is being used for habitat classification- more of an introduction to the context rather than jumping straight in to the technical.
Line 15: ‘ One of the parts of wetland monitoring is mapping vegetation’ Doesn’t sound quite right, how about ‘one aspect’?
Introduction
Line 34: Don’t need to say ‘ These areas ‘would say ‘they’
Lines 33-38 First paragraph- very short sentences, some of which could be linked, doesn’t flow very well.
Line 39 national park should be national parks
Lines 41-42
‘because it allows for a significant reduction amount of field measurements whether they enhance the objectivity and comparability of results’ should say ‘reduction in the amount’ I’m not quite sure what you mean by this sentence- do you mean you can reduce the number of field measurements regardless of whether it improves the objectivity and comparability of results?
Line 43- algorithms allow to combine and processed of different types of data like optical and ALS Doesn’t make sense perhaps ‘algorithms allow combination and processing of…’
Line 53 ‘may be not enough precise’ may not be precise enough
Lines 94-95 ‘A few studies would assess the usefulness of TI for identifying natural and semi-natural non-forest communities, including wetlands’ not sure what you are trying to say here probably because language not quite right.
Methods
Line 128 In 1984 research area was 128 included ‘the research area’
Line 135 should be The park.
Good description of the study site
Detailed description of image processing.
Line 222 language- collect a set representing the entire communities variability in the study area’. ‘collect a set representing the variability of all of the communities?’
Good use of ground survey data- seems comprehensive.
Very thorough testing of different data sources and metrics.
For those less familiar with the method and terminology would be good to include a definition/description of the different TI metrics, I cant see them in the supplementary. Also some of the other ‘descriptions’ in Table A2 aren’t that helpful and just repeat the measure. Some additional explanation would be desirable.
I’m not quite clear how the ground survey data were used to validate the classifications.
Figure 9 is very good.
Discussion
Line 487- needs re-writing ‘Based on conducted studies, the most useful ?? to differentiate wetland communities classes were mean TI calculated on MNF, SI, ALSF and TOPO’
Line 518 typo ‘drown’
Good description of the study site
Detailed description of image processing.
It would be good if the discussion could contain a bit more about the implementation of the findings to wetland conservation. It is a very technical paper with detailed descriptions of the methods, but the consequences and applications could be better discussed.
The language needs moderate revision. There are many grammatical errors and some spelling mistakes.
Reviewer 3 Report
The paper is very well done and the results are well presented. The only thing that could be considered would be to shorten and simplify the methodological part, which is sometimes too detailed. I have nothing further to add to the article.
Reviewer 4 Report
In this paper, LiDAR and hyperspectral data are utilized for mapping wetland vegetation in Warta River Mouth National Park, Poland. The paper recognizes the limitations of spatial data in areas with high heterogeneity and provides suggestions for future research and improvements. Overall, the paper is well-written. I would however recommend that authors should provide a dedicated section in the main methodology section explaining how the accuracy assessment methods were performed in this study.
Minor editing of English language required
